# Planting Density and Fertilization Evidently Influence the Fiber Yield of Hemp (*Cannabis sativa* L.)

**Gang Deng** [1] **, Guanghui Du** [1]**, Yang Yang** [1]**, Yaning Bao** [2] **and Feihu Liu** [1,*]

[1] School of Agriculture, Yunnan University, Kunming 650504, China
[2] College of Plant Science and Technology, Huazhong Agricultural University, Wuhan 430070, China
* Correspondence: xrhynu@hotmail.com; Tel.: +86-871-65031539; Fax: +86-871-6503-1539

**Abstract:** Hemp is one of the most important green (i.e., environmentally sustainable) fibers. Planting density, nitrogen (N), phosphorus (P) and potassium (K) significantly affect the yield of hemp fiber. By optimizing the above main four cultivation factors is an important way to achieve sustainable development of high-fiber yield hemp crops. In this study, the effects of individual factors and factor × factor interactions on the yield of hemp fiber over two trial years were investigated by the central composite design with four factors, namely planting density, nitrogen application, phosphorus application, and potassium application rate. The influences of these four test factors on the yield of hemp fibers were in the order nitrogen fertilizer ($X_2$) > planting density ($X_1$) > potassium fertilizer ($X_4$) > phosphate fertilizer ($X_3$). To obtain yields of hemp with high-quality fiber greater than 2200 kg ha$^{-1}$, the optimal range of cultivation conditions were planting density 329,950–371,500 plants/ha, nitrogen application rate 251–273 kg ha$^{-1}$, phosphorus application rate 85–95 kg ha$^{-1}$, and potassium application rate 212–238 kg ha$^{-1}$. This study can provide important technical and theoretical support for the high-yield cultivation of hemp fiber into the future.

**Keywords:** planting density; fertilization; the central composite design; fiber yield; analog optimization

---

## 1. Introduction

Hemp (*Cannabis sativa* L.) is an ancient and eco-friendly cultivated crop that was first cultivated in China, and is currently used for the manufacture of clothes, household supplies, paper pulp, drugs, food, and recyclable composite materials and among others [1–3]; hemp is known to have been used to make more than 2500 products [4,5]. Hemp textile industries first began in Europe and Asia around 8000 BC [6]. In the middle of the 20th century, hemp was banned from cultivation by governments as an illegal drug crop. However, in recent years, governments and researchers became more interested in the cultivation of hemp, as one of the most important crops for green fiber, seed oil (rich in omega-3 and omega-6 in the right ratio) and domestic drugs uses [2,5]. The cultivation of a number of hemp cultivars with low (<0.3%) THC (tetrahydrocannabinol) concentration has been allowed, to the point where some European governments even provide agricultural subsidies for hemp cultivation [7]. Hemp can be grown with little or no chemical fertilizers, herbicides or pesticides and the crop is now cultivated all around the world [6].

The leading plant macronutrients, nitrogen (N), phosphorus (P) and potassium (K), are important components of plant amino acids, hormones, genetic materials (DNA and RNA) and other substances. Also, it involved in many life processes, such as plant growth, metabolism, cell structure, signal transduction, osmotic regulation, and response to stresses [8–10]. However, due to the low efficiency of utilization of crop fertilizers, especially NPK fertilizers, more than 50% of the nutrients applied to land as chemical fertilizers is wasted [11,12], which also leads to contamination of soil and

water resources [13]. Therefore, it is important to determine how to effectively utilize fertilizers or to improve the nutrient-use efficiency of crops to achieve high fiber yields while protecting the environment. The characteristics of hemp are rapid growth, a well-developed root system and a high above-ground biomass (25,000 kg ha$^{-1}$) [14]. Its growth is sensitive to environmental factors, particularly fertilizers. Of these, the demand for nitrogen by fiber type hemp is greater than that for phosphate or potassium [15,16]. An appropriate planting density allows for efficient use of available resources, such as light, water and nutrients, by the crop, significantly increasing yield of hemp fibers [17,18]. In general, a high planting density is associated with the production of high-quality long fibers [19]. However, different varieties in different regions vary in their optimal planting densities, although an appropriate planting density for hemp fiber cultivation in China is 40–60 plant m$^{-2}$ [20–23].

According to current research, both NPK application rates and planting density can influence the fiber yield of hemp. However, the extent of their respective impacts on the yield of hemp for the fiber industry, the pattern of the impacts, and the optimal cultivation methods are still unclear. In addition, due to the unscientific use of fertilizer in actual production, it not only causes waste of fertilizer, but also leads to an increase in production costs. Therefore, studying the effects of fertilizers and density on the fiber yield of hemp can effectively solve these problems. For example, this study can provide a suitable NPK ratio, optimal planting density, etc.

In order to identify the optimal agronomic conditions (including planting density, nitrogen, phosphorus, potassium) for high-yield cultivation of hemp fiber, the current study analyzed the extent of the effects of the individual factors N, P and K fertilizer application rates and planting density on the yield of hemp fibers, using the most important hemp variety in China, 'Yunma 1' as the test material. Varies optical agronomic methods can be obtained in this study, and it will provide important technical and theoretical support for the high-yield cultivation of hemp fiber into the future.

## 2. Materials and Methods

### 2.1. Materials

The experiment was carried out on the experimental farm of the Agricultural College of Yunnan University, Kunming, China, in 2016 and 2017, on sites with uniform soil fertility. The basic soil characteristics were pH 5.98, organic matter content 35.85 g kg$^{-1}$, total nitrogen 0.17%, total phosphorus 0.09%, total potassium 1.63%, available nitrogen 151.3 mg kg$^{-1}$, available phosphorus 44.04 mg kg$^{-1}$, and available potassium 239 mg kg$^{-1}$. The farm was unprecedented, with good irrigation and drainage conditions. Kunming lies at 25°01′ E, 102°41′ N, at an altitude of 1896 m, and it has a dry season from November to April, with an annual rainfall of 2016, 2017 were 1017 mm, 1049 mm, respectively, and a monthly average temperature of 2016, 2017 were 15.6 °C, 16 °C, 1049 mm, respectively. The test material was 'Yunma 1' (THC < 0.3%), a fiber hemp variety, seeds of which were provided by the Yunnan Academy of Science.

### 2.2. Methods

Four factors, planting density ($X_1$), nitrogen fertilizer rate ($X_2$), phosphate fertilizer rate ($X_3$), and potassium fertilizer rate ($X_4$), were tested in this study. Five levels were set for each factor. The experiment was conducted in each year by the central composite design with four factors and 36 combinations. The dimensions of each trial plot were 3.5 m × 2.6 m (area 9.1 m$^2$), with the 36 plots set out in a completely randomized arrangement, and all the 36 combinations with three replicates (total 108 plots).

The fertilizers applied in this study were urea (containing 46% N), calcium phosphate (containing 14% P$_2$O$_5$), and potassium chloride (containing 54% K$_2$O). Table 1 shows the variable factors and their levels. All phosphate fertilizer and potassium fertilizer were incorporated into the seedbed as base fertilizer, and nitrogen fertilizer was applied twice in March (60%, sowing), June (40%, rapid growth period). The seeds were sown by hand in early May, with an inter-row spacing of 40 cm, and a total

of eight rows of hemp are planted in each plot. No chemical pesticides were used during the entire growth period. The hemp was harvested when mature (late September, 70–80% male plant flowering). Twenty plants in the middle of the plot (chosen from rows No. 3 to 5) were randomly selected in each plot. The rods and fibers in the above-ground biomass were separated using a special stripping machine. The fibers were dried (80 °C) and weighed. The yield of fibers (kg ha$^{-1}$) per plot was then calculated according to the effective number of plants in each plot.

**Table 1.** Central composite design of plant density and fertilizer dose.

| Agronomic Variable | Alternative Gradient | Variable Design | | | | |
|---|---|---|---|---|---|---|
| | | **−2** | **−1** | **0** | **1** | **2** |
| Density (plants ha$^{-1}$) ($X_1$) | 150 000 | 100,000 | 250,000 | 400,000 | 550,000 | 700,000 |
| N (kg ha$^{-1}$) ($X_2$) | 75 | 75 | 150 | 225 | 300 | 375 |
| P (kg ha$^{-1}$) ($X_3$) | 30 | 30 | 60 | 90 | 120 | 150 |
| K (kg ha$^{-1}$) ($X_4$) | 75 | 75 | 150 | 225 | 300 | 375 |

*2.3. Data Analysis*

The experimental design was based on the calculation principle of the combined design of the central composite design [24]. This study used the software processing system of the statistical software DPS 6.01 (Hangzhou Ruifeng Information Technology Co., Ltd., Hangzhou, China) [25], to establish the mathematical model of indicators such as fiber yield (dependent variables) and test factors (independent variables), and to statistically analyze the model. In this paper, *p*-value < 0.05 was used as a significant difference level, but the difference is a not significant difference.

## 3. Results

*3.1. Establishment and Verification of the Fiber Yield Model*

The fiber yield of hemp under different combinations ranged from 1701 to 3205 kg ha$^{-1}$ (Table 2). The results were analyzed using the regression model. With the yield of hemp fibers as the target trait (*Y*), the regression model was established between the test factors ($X_1$, $X_2$, $X_3$ and $X_4$) and the target trait:

$$Y = 2907.75 - 112.75X_1 - 156.17X_2 + 1.08X_3 - 21.50X_4 - 196.02X_1{}^2 - 160.15X_2{}^2 - 123.65X_3{}^2 - 144.15X_4{}^2 + 28.631X_2 - 2.25X_1X_3 - 65.88X_1X_4 - 70.75X_2X_3 + 46.63X_2X_4 - 9.00X_3X_4 \quad (1)$$

Through the analysis of variance (ANOVA) of fiber yield (Table 3), the best fitting models were determined by multiple linear regressions with backward elimination, whereby non-significant (*p* > 0.05) factors and interactions were removed from models. The determination coefficient for hemp fiber yield in this study was r$^2$ = 0.699, meaning that the models explained 70% of the variability in hemp fiber yield and were found to be adequate for the data. Analysis of variance (ANOVA) also showed that the regression models for hemp fiber yield were significant, and the models had no significant lack of fit (0.499, *p* > 0.05) (Table 2). In this way, well-fitting models for hemp fiber yield were established. Not all interaction parameters were significant (*p* > 0.05) (Table 3).

**Table 2.** Structure matrix and the production results from 2016 and 2017.

| No | Density ($X_1$) | N ($X_2$) | P$_2$O$_5$ ($X_3$) | K$_2$O ($X_4$) | Mean Fiber Yield (kg ha$^{-1}$) |
|----|------|------|------|------|------|
| 1 | 1 | 1 | 1 | 1 | 2251 |
| 2 | 1 | 1 | 1 | −1 | 2479 |
| 3 | 1 | 1 | −1 | 1 | 2193 |
| 4 | 1 | 1 | −1 | −1 | 2821 |
| 5 | 1 | −1 | 1 | 1 | 2051 |
| 6 | 1 | −1 | 1 | −1 | 2131 |
| 7 | 1 | −1 | −1 | 1 | 1708 |
| 8 | 1 | −1 | −1 | −1 | 2239 |
| 9 | −1 | 1 | 1 | 1 | 2387 |
| 10 | −1 | 1 | 1 | −1 | 2523 |
| 11 | −1 | 1 | −1 | 1 | 2824 |
| 12 | −1 | 1 | −1 | −1 | 2399 |
| 13 | −1 | −1 | 1 | 1 | 2035 |
| 14 | −1 | −1 | 1 | −1 | 2603 |
| 15 | v | −1 | −1 | 1 | 2102 |
| 16 | −1 | −1 | −1 | −1 | 2236 |
| 17 | −2 | 0 | 0 | 0 | 2436 |
| 18 | 2 | 0 | 0 | 0 | 1701 |
| 19 | 0 | −2 | 0 | 0 | 1968 |
| 20 | 0 | 2 | 0 | 0 | 2456 |
| 21 | 0 | 0 | −2 | 0 | 2336 |
| 22 | 0 | 0 | 2 | 0 | 2380 |
| 23 | 0 | 0 | 0 | −2 | 1935 |
| 24 | 0 | 0 | 0 | 2 | 2617 |
| 25 | 0 | 0 | 0 | 0 | 2833 |
| 26 | 0 | 0 | 0 | 0 | 3175 |
| 27 | 0 | 0 | 0 | 0 | 3048 |
| 28 | 0 | 0 | 0 | 0 | 2976 |
| 29 | 0 | 0 | 0 | 0 | 3294 |
| 30 | 0 | 0 | 0 | 0 | 3000 |
| 31 | 0 | 0 | 0 | 0 | 2849 |
| 32 | 0 | 0 | 0 | 0 | 2809 |
| 33 | 0 | 0 | 0 | 0 | 2936 |
| 34 | 0 | 0 | 0 | 0 | 3205 |
| 35 | 0 | 0 | 0 | 0 | 2579 |
| 36 | 0 | 0 | 0 | 0 | 2189 |

**Table 3.** Analysis of variance (ANOVA) of fiber yield of hemp.

| Source | Sum of Squares | *df* | Mean Squares | Partial Correlation | *F*-Value | *p*-Value |
|--------|------|------|------|------|------|------|
| $X_1$ | 305,101.49 | 1 | 305,101.49 | −0.3761 | 3.4596 | 0.0770 |
| $X_2$ | 585,312.64 | 1 | 585,312.64 | 0.4900 | 6.6369 | 0.0176 * |
| $X_3$ | 28.17 | 1 | 28.17 | 0.0039 | 0.0003 | 0.9859 |
| $X_4$ | 11,094.00 | 1 | 11,094.00 | −0.0772 | 0.1258 | 0.7264 |
| $X_1^2$ | 122,9573.30 | 1 | 1,229,573.30 | −0.6317 | 13.9422 | 0.0012 ** |
| $X_2$ | 820,693.98 | 1 | 820,693.98 | −0.5541 | 9.3059 | 0.0061 ** |
| $X_3^2$ | 489,225.33 | 1 | 489,225.33 | −0.4571 | 5.5474 | 0.0283 * |
| $X_4^2$ | 664,896.66 | 1 | 664,896.66 | −0.5140 | 7.5393 | 0.0121 * |
| $X_1X_2$ | 13,110.25 | 1 | 13,110.25 | 0.0838 | 0.1487 | 0.7037 |
| $X_1X_3$ | 81.00 | 1 | 81.00 | −0.0066 | 0.0009 | 0.9761 |
| $X_1X_4$ | 69,432.25 | 1 | 69,432.25 | −0.1901 | 0.7873 | 0.3850 |
| $X_2X_3$ | 80,089.00 | 1 | 80,089.00 | −0.2036 | 0.9081 | 0.3515 |
| $X_2X_4$ | 34,782.25 | 1 | 34,782.25 | 0.1358 | 0.3944 | 0.5368 |
| $X_3X_4$ | 1296.00 | 1 | 1296.00 | −0.0264 | 0.0147 | 0.9047 |
| Regression | 4,304,716.47 | 14 | 307,479.75 | F2 = 3.48654 | | 0.0100 ** |
| Residual | 1,851,999.75 | 21 | 88,190.46 | | | |
| Lack of fit | 865,925.50 | 10 | 86,592.55 | F1 = 0.96597 | | 0.4991 |
| Pure error | 986,074.250 | 11 | 89,643.11 | | | |
| Total error | 6,156,716.2222 | 35 | | | | |

Note: * $p < 0.05$, ** $p < 0.01$. *df*, degree of freedom, $X_1$, planting density, $X_2$, nitrogen, $X_3$, phosphate, $X_4$, potassium.

### 3.2. Main-Effect Analysis of Factors

The sub-models of the relationship between hemp fiber yield and the main effects of planting density, nitrogen fertilizer, phosphorus fertilizer and potassium fertilizer were calculated by using the method of descending dimension as the mathematical model with the other factors of the fixed Equation # 11 at 0 levels (Equation (2)):

$$
\begin{aligned}
&\text{Planting density: } Y_1 = 2907.75 - 112.75X_1 - 196.02X_1{}^2 \\
&\text{Nitrogen fertilizer: } Y_2 = 2907.75 + 156.17X_2 - 160.15X_2{}^2 \\
&\text{Phosphorus fertilizer: } Y_3 = 2907.75 + 1.\ 08X_3 - 123.65X_3{}^2 \\
&\text{Potassium fertilizer: } Y_4 = 2907.75 - 21.50X_4 - 144.15X_4
\end{aligned}
\tag{2}
$$

The above equation showed that the partial regression coefficients of planting density ($X_1$), nitrogen ($X_2$), phosphorus ($X_3$) and potassium application rates ($X_4$) were −112.75, 156.17, 1.08 and −21.5, respectively. As positive effects, increasing nitrogen and phosphorus would increase the yield of hemp fiber, whereas, as negative effects, increasing planting density and potassium as would reduce the yield of hemp fiber. According to the absolute value discriminant method of linear coefficients, the influence of each factor on fiber yield can be defined directly from the absolute value of the respective regression coefficient, as in the order nitrogen > planting density > potassium > phosphorus.

### 3.3. Analysis of Single Factor Effects

Figure 1 shows that, according to the sub-model (Equation (2)), the four test factors have a parabolic relationship with hemp fiber yield within the constraint range of −2 ≤ Xi ≤ 2. Fiber yield increased with increasing plant density level from −2 to 0.29, and then decreased, with a maximum fiber yield of 2923 kg ha$^{-1}$ (at −0.29; 356,500 plants ha$^{-1}$). Fiber yield increased with increasing nitrogen level from −2 to 0.49, then decreased, with a maximum fiber yield of 2946 kg ha$^{-1}$ (at 0.49; 262 kg ha$^{-1}$). Fiber yield increased with increasing phosphorus level from -2 to 0, and then decreased, with a maximum fiber yield of 2908 kg ha$^{-1}$ (at 0; 90 kg ha$^{-1}$). Fiber yield increased with increasing potassium level from −2 to 0.07, and then decreased, with a maximum fiber yield of 2906 kg ha$^{-1}$ (at 0.07; 230 kg ha$^{-1}$). Figure 1 illustrates that fiber yield decreased rapidly at planting density and potassium levels from 0–2, the decrease being faster than with nitrogen fertilizer and phosphate fertilizer, with planting density decreasing the fastest, and nitrogen decreasing at the slowest rate. Therefore, the fiber yield of hemp would not increase but would decrease rapidly once the plant density and potassium fertilizer levels increased beyond the optimal level.

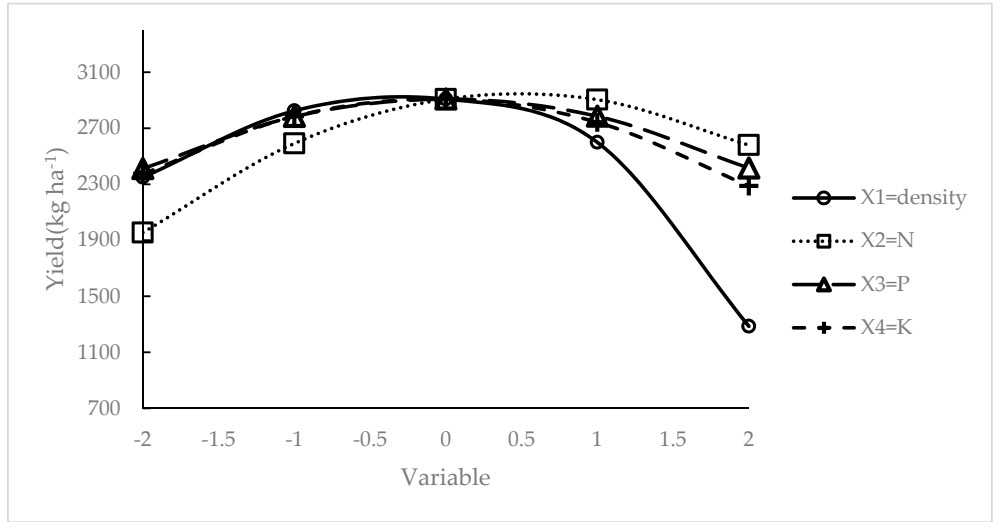

**Figure 1.** The effect of the factors ($X_1$ = density, $X_2$ = N, $X_3$ = P, $X_4$ = K) on mean fiber yield for 2016 and 2017.

### 3.4. Analysis of the Marginal Yield Effect of Single Factors

Marginal yield is the increase in yield for each additional unit of a variable factors. The rate of change caused by each factor could be calculated by the first-order partial derivative ($dY/dX_i$) of the fiber yield ($Y$) in response to a particular factor ($X_i$). It was further analyzed to obtain the reasonable collocation of the levels of different factors when a certain factor was the main control factor, and the highest yield under different conditions. The sub-model (Equation (3)) of the marginal yield effect of a single factor was calculated from the first derivative in the sub-model (Equation (2)):

$$\text{Planting density: } dY/dX_1 = -112.75 - 392.04X_1$$
$$\text{Nitrogen fertilizer: } dY/dX_2 = 156.17 - 320.3X_2$$
$$\text{Phosphorus fertilizer: } dY/dX_3 = 1.08 - 247.3X_3 \quad (3)$$
$$\text{Potassium fertilizer: } dY/dX_4 = -21.50 - 288.3X_4$$

According to the corresponding marginal models with different horizontal values of each factor, the planting density showed the greatest change in the marginal effect changes at different levels of each factor, while nitrogen and potassium showed less change, and phosphorus showed the least change. The marginal effects of all four factors, from 0 level to the highest level, were negative, which indicated that the increase in NPK application rate and planting density led to reduced hemp fiber yield, which also indicated that reduction in fertilizer application or planting density could increase the fiber yield of hemp. However, combining the fiber yield regression model, it was clear that there were certain interaction effects among the factors, but that the difference in the interaction effect between each factor was not significant. The influence of each factor on the increase of fiber yield at different levels varied, and the order of their effects on fiber yield increase was: $X_2 > X_1 > X_4 > X_3$ at the $-2$ and $-1$ levels; the order of their effects on fiber yield reduction was $X_1 > X_4 > X_3 > X_2$ at the 1 and 2 levels (Figure 2).

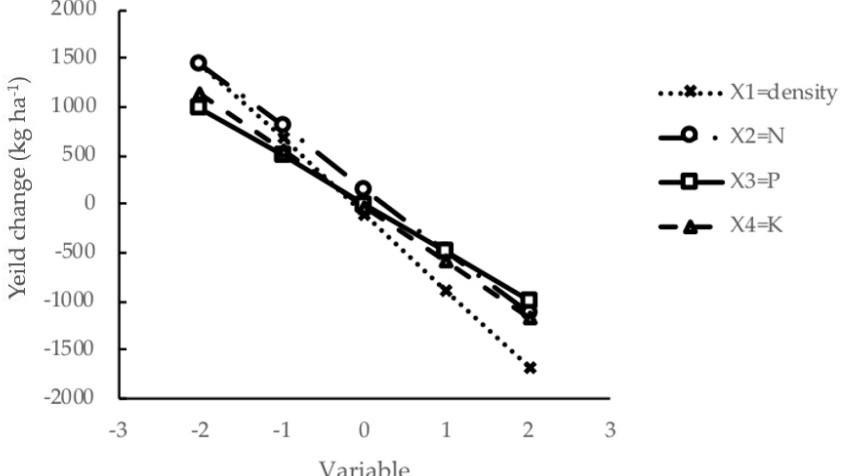

**Figure 2.** Marginal effects of increases in experimental factors ($X_1$ = density, $X_2$ = N, $X_3$ = P, $X_4$ = K) on boundary fiber hemp production.

### 3.5. Optimization of Agronomic Methods Plan

Within the constraint range of $-2 \leq X_i \leq 2$, 134 sets of combinations with the yield of hemp fiber greater than 2200 kg ha$^{-1}$ were selected and further analyzed for frequency (Table 4). Table 4 shows that there were many routes by which to obtain high fiber yield according to various agronomic methods as shown by the combination plans with high yields of hemp fiber but were mainly concentrated in the horizontal $-1$ to $+1$ level range.

**Table 4.** Frequency of special $X_i$ value of hemp fiber yield beyond 2200 kg ha$^{-1}$.

| Factor | | Density ($X_1$) | | N ($X_2$) | | P ($X_3$) | | K ($X_4$) | |
|---|---|---|---|---|---|---|---|---|---|
| | | Degree | Frequency (%) | Degree | Frequency (%) | Degree | Frequency (%) | Degree | Frequency (%) |
| Variable design | −2 | 10 | 7.46 | 0 | 0 | 8 | 5.97 | 8 | 5.97 |
| | −1 | 44 | 32.84 | 19 | 14.18 | 36 | 26.87 | 36 | 26.87 |
| | 0 | 60 | 44.78 | 48 | 35.82 | 46 | 34.33 | 46 | 34.33 |
| | 1 | 20 | 14.93 | 48 | 35.82 | 36 | 26.87 | 36 | 26.87 |
| | 2 | 0 | 0 | 19 | 14.18 | 8 | 5.97 | 8 | 5.97 |
| Weighted mean | | −0.328 | | 0.500 | | 0 | | 0 | |
| Standard error | | 0.0710 | | 0.0780 | | 0.0870 | | 0.0870 | |
| 95% Confidence interval | | −0.467~−0.190 | | 0.347~0.653 | | −0.171~0.171 | | −0.171~0.171 | |
| Optimal range | | 329,950~371,500 plant ha$^{-1}$ | | 251.03~273.98 kg ha$^{-1}$ | | 84.87~95.13 kg ha$^{-1}$ | | 212.18~237.83 kg ha$^{-1}$ | |

Under similar experimental conditions, in order to obtain high yields of raw hemp fiber greater than 2200 kg ha$^{-1}$, the relatively optimal combination plan of cultivation involved a planting density of 329,950–37,1500 plants ha$^{-1}$, a nitrogen application rate of 251–273 kg ha$^{-1}$, a phosphorus application rate of 85–95 kg ha$^{-1}$, and a potassium application rate of 212–238 kg ha$^{-1}$.

## 4. Discussion

Hemp has attracted much attention from the market and from researchers due to its multiple uses, therefore, improving the production of hemp fiber through research will help promote the promotion and competitiveness of hemp products [26,27]. Many studies have focused on the effects of N fertilizer application rate and planting density on the growth and fiber yield of hemp [15,18,28], but the current study is the first to provide a comprehensive analysis of the effects and interactions of NPK fertilizer application rates and planting density on hemp fiber yield. Because hemp for the fiber industry has the characteristics of high biomass and rapid growth, a large amount of fertilizer is required during the growth period. It has been reported that the amount of NPK applied and hemp fiber yield per unit area were positively correlated [29]. Nitrogen fertilizer in the current study exhibited the greatest influence on the fiber yield of hemp, and its contribution to yield at high nitrogen level was higher than that of the other three factors; at a moderate nitrogen level (level 0; 225 kg ha$^{-1}$), however, nitrogen contributed less to fiber yield than did the other three factors. When nitrogen level reached 262 kg ha$^{-1}$, the fiber yield was maximal, with yield response to nitrogen application rate increasing more at −2 to 0 levels than that at the 0 to 2 levels. The results were similar to those reported by Struck et al. [14]. The current study also found that the fiber yield at the high nitrogen level 2 was still higher than that at the lowest nitrogen level −2, while the fiber yield at the highest level of the other treatments was lower than that at the lowest level, an observation which demonstrated the importance of optimizing nitrogen fertilizer application to achieving the goal of high hemp fiber yield.

This present study revealed that increasing either phosphorus or nitrogen application rates exhibited a positive effect on hemp fiber yield. However, the effect of phosphorus on hemp fiber yield was smaller, the increase was not significant, and the overall change curve was relatively flat, findings which were similar to those reported by Vera et al. [30,31]. Meanwhile, the current study found that increasing either potassium application rate or planting density exhibited a negative effect on hemp fiber yield, results which differed from those of previous studies that showed increased hemp fiber yield in response to increased potassium, but were in line with the results of Finnan and Burke's research [32], which concluded that there was no significant correlation between hemp fiber yield and soil potassium levels. The demand for potassium by hemp may be lower than expected. Despite there being high potassium uptake by hemp under high-potassium conditions, the extra uptake of

potassium had no significant effect on fiber yield increase of hemp, which was considered to be luxury uptake [32].

Planting density directly affects the structure of the hemp population, and thus affects the fiber yield. According to previous studies, it has found that, when the density reached a certain level, hemp fiber yield decreased due to a self-thinning effect [28]. In this study, it was found that increasing the planting density had a negative effect on the yield of hemp. The fiber yield level was lower than that achieved by other factors at planting density levels above 0, and the decrease was the greatest. Therefore, it is not appropriate to increase the planting density in hemp production, as, once the planting density exceeded a certain range, fiber yield per area was significantly reduced. The present study demonstrated that the optimal planting density was 32–37 plants m$^{-2}$.

In order to obtain high fiber yield of hemp under similar conditions to those experienced in the present study, this study optimized the agronomic methods, and showed the relatively optimal combination plan of cultivation methods which could reach high fiber yields of greater than 2200 kg ha$^{-1}$, namely planting density of 329,950–371,500 plants/ha, a nitrogen application rate of 251–273 kg ha$^{-1}$, a phosphorus application rate of 85–95 kg ha$^{-1}$, and potassium application rate of 212–238 kg ha$^{-1}$, with an approximate N:P:K fertilizer application ratio (relative to the soil NPK levels described in Section 2.1) of 5:2:4. This present study can provide important guidance for optimizing the agronomic conditions for hemp cultivation for fiber.

## 5. Conclusions

The four tested factors effects on the fiber yield of hemp was shown in this study to be in the order nitrogen fertilizer rate ($X_2$) > planting density ($X_1$) > potassium fertilizer rate ($X_4$) > phosphate fertilizer rate ($X_3$). The study also revealed that increasing the amount of N, P, or K applied or the planting density could lead to fiber yield reductions. This study suggested that the relatively optimal combination plan of cultivation to obtain hemp fiber yield greater than 2200 kg ha$^{-1}$ involved a planting density of 329,950–371,500 plants ha$^{-1}$, a nitrogen application rate of 251–273 kg ha$^{-1}$, a phosphorus application rate of 85–95 kg ha$^{-1}$, and a potassium application of 212–238 kg ha$^{-1}$, with an approximate N: P: K fertilizer.

**Author Contributions:** Conceptualization, F.L; methodology, G.D. (Gang Deng); resources, Y.Y.; data curation, G.D. (Guanghui Du); writing—original draft preparation, G.D. (Gang Deng); writing-review & editing, F.L., Y.B.; funding acquisition, F.L.

**Funding:** This research was supported by a grant from the Natural Science Foundation of Yunnan (2016FB068 and 2017FD060), and China Agriculture Research System (CARS-16-E15).

**Conflicts of Interest:** The authors declare no conflict of interest.

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
