# Peer review of "Planting Density and Fertilization Evidently Influence the Fiber Yield of Hemp (Cannabis sativa L.)"

_agronomy, doi:10.3390/agronomy9070368_

Reviewer 1 Report

Dear Editor and Authors,

First of all I would like to thank you for considering my-self for reviewing your manuscript.

The manuscript fit with the aims and scope of the journal. It contribute on the discussion regarding the agronomical practice of industrial hemp. In the recent period, this crop has been reintroduced in different cropping system worldwide and there is a great interest for the possible production that could be obtained, especially in seed and inflorescence production. The fiber production is less important due to the strong competition with synthetic fiber. However, it is interesting because it is a sustainable way to produce textile fibers.

It is commonly accepted that the cultivation of industrial hemp not require a lot of external input expecially pesticides and fertilizers, in any case for high yield in terms of quality and quantity more attention is needed to adrees the crop production level.

In my opinion the manuscript, it of interest and it merits to be published, even if there are some point that should be clarified in order to make the manuscript more understandable for readers. For this reason at this step I suggest major revision.

Detailed suggestions regarding the manuscript are reported in the attached pdf file.

thank you again for giving me the opportunity to review this manuscript

best regards.

Author Response

Response to Reviewer 1 Comments

Point 1: The title is too generic, I suggest to rewrite the title more concise and informative. Titles are often used in information-retrieval systems. Avoid title with effect of.... impact of.....

Response 1: Revised as suggested. The title has been changed as “, please see the abstract, page 1 line 22. Planting density and fertilization evidently influence the fiber yield of hemp (Cannabis sativa L.)”

Point 2: This sentence is not clear. Please rewrite in order to highlight why industrial hemp should be cultivated for fiber production. What is the main cultuvation factors? please specify..

Response 2: Revised as suggested. We have been rewritten it as “ Planting density, nitrogen (N), phosphorus (P) and potassium (K) significantly affect the yield of hemp fiber. By optimizing the above main four cultivation factors is an important way to achieve sustainable development of high-yield hemp crops.” Please see page 15-17.

Point 3: Nowadays, industrial hemp is a crop cultivated not only for fiber production but also for seed and inflorescence production, thus yield is generic term, I prefer to use here and through the text another term (i.e. fiber yield).

Response 3: Revised as suggested. The yield has been changed as “fiber yield”

Point 4: kg ha-1,Please coorect the unit of measure here and through the manuscript.

Response 4:  Revised as suggested. We have been changed the “kg/ha, plant/m2, g/kg, etc.” as “kg ha-1, plant m-2, g kg-1 , etc.”

Point 5: This key words is already in the title, please substitute with another more actractive.

Response 5: Revised as suggested.  We have been change the key to “Planting density,fertilization”

Point 6: In my opinion, the renewed interest on industrial hemp is related to its seed production that can be used for oil (righ in omega-3 and omega-6 in the right ratio) protein for vegans. Moreover another production is related to essential oil and pharmaceutical products. Limit your sentence to domestic drugs is too reductive.

Response 6:  Revised as suggested. We have been rewritten it as “as one of the most important crops for green fiber, seed oil (rich in omega-3 and omega-6 in the right ratio) and domestic drugs uses.”  Please see page 1, line 37-38.

Point 7: Really it depends on the products that want to be obtained, i.e. nitrogen is important for seed production, phosphorous is important for fiber quality.... please try to improve this sentence with more information.

Response 7: Revised as suggested. We have been rewritten it as “Of these, the demand for nitrogen by fiber type hemp is greater than that for phosphate or potassium” Please see page 2, line 53-54.

Point 8: Please add the hypothesis of the manuscript.

Response 8:  Revised as suggested. We have been added the hypothesis of the study as “For example, this study can provide a suitable NPK ratio, optimal planting density, etc.”  Please see page2 line 65-66.

Point 9: Please add the main objectives of the manuscript. at this step are not completely clear.

Response 9:  Revised as suggested.  We have been rewritten this paragraph. Please see page 2 line 67-72.

Point 10:  In each year?

Response 10:  Yeswe have been add it in our paper. Please see page 2 line 89.

Point 11:  is it the common practice adopted by the farmers in the area? In my opinion, P and K fertilizer shold be applied at sowing, while N due to its high mobility should be splitted at least in two times (sowing and before canopy closure. In any case please specify.

Response 11: Revised as suggested.  We have been rewritten it as “ All phosphate fertilizer and potassium fertilizer were incorporated into the seedbed as base fertilizer, and Nitrogen fertilizer was applied twice in March (60%, sowing), June (40%, rapid growth period).”  Please see page 3 line 95-97.

Point 12: What do you mean mature, the time of harvesting in hemp depend on the type of products (fiber, inflorescence and seed). Thus specify the phenological phase of industrial hemp?

Response 12: Revised as suggested.  We add the sentence “late September, 70-80% male plant flowering” in our paper. Please see page 3 line 99.

Point 13:  This sentence should be moved in the materials and methods in the data analysis.

Response 13: Revised as suggested. We have been deleted it. Please see page 3 line 121.

Point 14: Do you perform also ANOVA? it is not clear in M&M. please improve M&M based the analysis data that you performed.

Response 14:  Revised as suggested. Please see page 3 line 123.

Point 15: What mean X1, X2, .... and X4? the table should be clarified, even the symbol meaning is already specified in the text. Please add the meaning in the footnote of the table.

what mean df? please specify degree of freedom in the caption

Response 15:  Revised as suggested. We have been add it in the table 3. Please see the caption of the table 3.

Point 16This sentence is a repetition of introduction section, in my opinion it could be eliminated.

Response 16:  Revised as suggested. We have been rewritten it. Please see page 7 line 202- 204.

Point 17Do you detect in your experiment self-thinning of the crop?

Response 17: Thanks for your suggestion, because the research in this paper mainly studies the yield of hemp fiber, we have not written this results in this paper.

Point 18per plant or per area? and what about the quality of the fibers? generally increasing the planting density the fiber yield per plant decrease while fiber yield per area increase. the quality is generally better in the high planting density.

Response 18:  Revised as suggested. We have been rewritten added the “per area”  in our paper. Please see page 8 line 239.

Reviewer 2 Report

Review of the paper by Deng et al. “Effect of planting density and fertilizer application on fibre yield of industrial hemp (Cannabis sativa L.).

The focus of the global and regional policies on circular bioeconomy and sustainable development influences research on new bio-based products including those that can contribute to a broad substitution of petrochemical products. Hemp is a plant of growing interest of the industry due to its potential for high yielding of biomass and multiple uses. The main uses of hemp biomass is for fiber and seeds. The reviewed paper is on agrotechnical factors that can be optimized in the context of the high yield of fiber. There are considered four factors in cultivation: plant density and fertilization of N, P and K.

Remarks

Title

Cannabis with the low amount of THC is considered as and industrial form and is conventionally called as hemp thus in the title the industrial hemp can be written just as hemp, i.e.: Effect of planting density and fertilizer application on fibre yield of hemp (Cannabis sativa L.).”

Objectives

… most important … ?

Methods

In the research it was applied a very effective experimental design that enable to compare many factors at the cost of higher order interactions with the use of strongly reduced experimental units, here field plots. The design has also limitations and it would be purposeful to mention on them.

Two years in agrotechnical studies can limit the quality of final conclusions. No indication in paper on the climatic and location conditions in years, no indication on interaction years x factors. Average from two years was taken into analysis. This analysis is valid only when interactions between years and factors are insignificant.

There is a methodic inconsistence. It was mentioned that experiment was in three replications while in the field it was only a single replicated experiment and probably three replication was applied in analytical work in lab. I maybe wrong but it needs for clarification.

From the technical point of view when there is applied such large amounts of fertilizers and small experimental units, in order to avoid neighboring interferences the area of plots for harvest should be planned smaller. In the other way there is not possible to separate potential confounding of experimental factors with soil variability and results are biased.

Conclusion from statistical analysis should be done at the same significance level, in this case p<0.05. It can be only mentioned in the chapter on Methods and there is no need to provide the level at each statistical statement in the Results.

Results and discussion

In general there is a clear development of obtained results and a proper discussion with literature.

Considering agricultural aspects of high yielding of hemp it could be easily anticipated the importance of factors, i.e. N, density, K and P. but in the paper the importance of the factors was not confirmed statistically. What was the measure of importance?

To recapitulate, I recommend this paper for publication after considering the above in my opinion minor inconsistencies (even if only two years).

Author Response

Response to Reviewer 2 Comments

Point 1: Title

Cannabis with the low amount of THC is considered as and industrial form and is conventionally called as hemp thus in the title the industrial hemp can be written just as hemp, i.e.: Effect of planting density and fertilizer application on fibre yield of hemp (Cannabis sativa L.).”

Objectives

most important … ?

Response 1: Revised as suggested. We have been change the “industrial hemp” to “hemp” in our paper, and we have been change the title as “Planting density and fertilization evidently influence the fiber yield of hemp (Cannabis sativa L.)”

Point 2: Methods

In the research it was applied a very effective experimental design that enable to compare many factors at the cost of higher order interactions with the use of strongly reduced experimental units, here field plots. The design has also limitations and it would be purposeful to mention on them.

Two years in agro technical studies can limit the quality of final conclusions. No indication in paper on the climatic and location conditions in years, no indication on interaction years x factors. Average from two years was taken into analysis. This analysis is valid only when interactions between years and factors are insignificant.

Response 2: Thanks for your useful suggestion. We have already queried the climatic factors (such as temperature and rainfall) in 2016 and 2017, and found that the climate difference between the two years is not significant. We have been added the climate conditions in our paper, please see page 2 line 80-83.

Point 3: There is a methodical inconsistence. It was mentioned that experiment was in three replications while in the field it was only a single replicated experiment and probably three replication was applied in analytical work in lab. I may be wrong but it needs for clarification.

Response 3:  Thanks for your useful suggestion. We have been rewritten this sentence. Please see page 2, line 90-92.

Point 4: There is From the technical point of view when there is applied such large amounts of fertilizers and small experimental  units, in order to avoid neighboring interferences the area of plots for  harvest should be planned smaller. In the other way there is not possible to separate potential confounding of experimental factors with soil variability and results are biased.

Response 4:  Thanks for your useful suggestion. In fact, the sampling method of this study is based on what you are doing. We have been rewritten it in our paper, please see page 3, line 100-101.

Point 5: Conclusion from statistical analysis should be done at the same significance level, in this case p<0.05. It can be only mentioned in the chapter on Methods and there is no need to provide the level at each statistical statement in the Results.

Response 5:  Thanks for your useful suggestion. We have been modified it as your suggestion. Please see page 3, line 110-111.

Point 6: Results and discussion

In general there is a clear development of obtained results and a proper discussion with literature.

Considering agricultural aspects of high yielding of hemp it could be easily anticipated the importance of factors, i.e. N, density, K and P. but in the paper the importance of the factors      was not confirmed statistically. What was the measure of importance?

Response 6:  Thanks for your useful suggestion. From our study, if we want to achieve high yields of hemp fiber, we can choose the optimal combination plan (134 combinations). In addition, our study showed that the influences of these four test factors on the yield of hemp fibers were in the order nitrogen fertilizer (X2) > planting density (X1) > potassium fertilizer (Χ4) > phosphate fertilizer (X3).

Round  2

Reviewer 1 Report

Thank you for the manuscript.